# Performance and Capability Assessment in Surgical Subtask Automation

**DOI:** 10.3390/s22072501

**Published:** 2022-03-24

**Authors:** Tamás D. Nagy, Tamás Haidegger

**Affiliations:** 1Antal Bejczy Center for Intelligent Robotics, EKIK, Óbuda University, Bécsi út 96/B, 1034 Budapest, Hungary; haidegger@irob.uni-obuda.hu; 2Doctoral School of Applied Informatics and Applied Mathematics, Óbuda University, Bécsi út 96/B, 1034 Budapest, Hungary; 3Biomatics Institute, John von Neumann Faculty of Informatics, Óbuda University, Bécsi út 96/B, 1034 Budapest, Hungary; 4Austrian Center for Medical Innovation and Technology (ACMIT), Viktor-Kaplan-Straße 2/1, 2700 Wiener Neustadt, Austria

**Keywords:** RAMIS, partial automation, robot surgery validation, robot surgery benchmarking

## Abstract

Robot-Assisted Minimally Invasive Surgery (RAMIS) has reshaped the standard clinical practice during the past two decades. Many believe that the next big step in the advancement of RAMIS will be partial autonomy, which may reduce the fatigue and the cognitive load on the surgeon by performing the monotonous, time-consuming subtasks of the surgical procedure autonomously. Although serious research efforts are paid to this area worldwide, standard evaluation methods, metrics, or benchmarking techniques are still not formed. This article aims to fill the void in the research domain of surgical subtask automation by proposing standard methodologies for performance evaluation. For that purpose, a novel characterization model is presented for surgical automation. The current metrics for performance evaluation and comparison are overviewed and analyzed, and a workflow model is presented that can help researchers to identify and apply their choice of metrics. Existing systems and setups that serve or could serve as benchmarks are also introduced and the need for standard benchmarks in the field is articulated. Finally, the matter of Human–Machine Interface (HMI) quality, robustness, and the related legal and ethical issues are presented.

## 1. Introduction

The appearance and global spread of Robot-Assisted Minimally Invasive Surgery (RAMIS) induced a revolution in healthcare during the past two decades. The new approach extended the benefits of traditional Minimally Invasive Surgery (MIS) and offered lower risk of complications and faster recovery to patients [1,2,3]. The advanced endoscopic system and surgical instruments of RAMIS enabled the execution of more computer-integrated workflow models, improving the outcome of the interventions, as in the case of the nerve-sparing technique in radical prostatectomy [4,5]. RAMIS offers a number of benefits to surgeons, such as ergonomics, motion scaling, and 3D vision. Undoubtedly, the most successful RAMIS device is the da Vinci Surgical System (Intuitive Surgical Inc., Sunnyvale, CA, USA), with almost 7000 da Vinci units installed worldwide, which have performed over 10 million procedures to date. It has been over twenty years since the first generation robot (Figure 1) was cleared by the U.S. Food and Drug Administration, and today, the fourth generation—referring to the da Vinci Xi, X, and SP—is available, along with the research-enhanced version of the original system [6,7], and even the recent pandemic could not significantly disrupt this domain [8].

Many believe that the next step in the advancement of surgery will be subtask-level automation [9]. Automating monotonous and time consuming subtasks may decrease the cognitive load on the surgeon, who could then better focus on the more critical steps of the operation [10,11]. Currently, many research groups are working on this problem [12,13]; some groups chose to work in ex vivo (or rarely in vivo) [14,15] or realistic phantom environments [16], but simplified silicone phantoms are utilized mostly [15,17,18,19,20,21]. In the most recent years, the automation of simple surgical training exercises on rigid [22,23,24,25,26,27,28] or deformable [29,30] phantoms tends to receive increasing attention. Among all the training exercises, the automation of different versions of peg transfer is presented in the most significant number of studies [22,23,24,25,26,27,28,31], probably due to its simplicity, enabling to elaborate the basic principles and best algorithms for automation.

Regardless of how intensive the research of surgical subtask automation is, there is no consensus, so far, on the choice of evaluation metrics for the implementation level, thus, it is hard to compare those methods to each other, or even to the technique of human surgeons. Fontana et al. state the following on autonomous robotics: *“Within computer science, autonomous robotics takes the uneasy role of a discipline where the features of both systems (i.e., robots) and their operating environment (i.e., the physical world) conspire to make the application of the experimental scientific method most difficult.”* [32]. According to their study, the difficulties caused by the large factor of uncertainties often lead to a methodological problem. Namely, it is practically challenging to perform accurate experimentation, thus methodological soundness often takes a secondary role in robotic studies, compromising repeatability and reproducibility of the research projects. The absence of those aspects forces even the best research works into the category of “proof of concept”, while on the other hand medical devices are among the most heavily regulated domains (as described in Section 7).

A contextual characterization model for surgical automation was proposed and an overview of performance metrics, benchmarks, robustness, ethical and legal questions were presented in the authors’ related paper [33]. In this article, this contextual characterization model for surgical automation is refined. The topics of performance metrics, benchmarks, robustness, ethics, standardization, and regulation are discussed and analysed more thoroughly, and methodlogical recommendations are formed. The context was also extended to questions regarding Human–Machine Interfaces (HMIs) and hand-overs.

This article is structured as follows. Section 2 discusses the characterization of autonomous surgical systems, and a characterization model is proposed. In Section 3, the candidates for standard performance metrics of surgical subtask automation are presented, scored, and evaluated, and a recommendation is formed on the choice of standard validation metrics. It is also discussed which of those metrics are usable to compare autonomous surgical subtask execution to the performance of human surgeons and to compare those autonomous methods to each other. Section 4 overviews the benchmarking environments usable to further standardize the validation of the autonomous surgical systems, and a list of candidates is proposed. In Section 5, the issues regarding the interface between the surgeon and the autonomous system and the problems with hand-overs are addressed. Section 6 discusses the importance of robustness and approaches to its measurement. Section 7 considers the legal and ethical questions of autonomous surgical systems. Finally, Section 8 concludes the topics addressed in the article.

## 2. Characterization of Autonomy

Before the discussion of the evaluation metrics takes place, it is important to define the contextual classification for the automation of surgical tasks. The Autonomy Levels for Unmanned Systems (ALFUS) Ad Hoc Workgroup of National Institute of Standards and Technology paid a significant effort to define a framework for the characterization of Unmanned Systems (UMS) from the aspect of autonomy. The resulted ALFUS Framework concerns a broad spectrum of UMSs, including unmanned aerial, ground, maritime vehicles, and unattended ground sensors which are applicable in areas like military, manufacturing, search and rescue, or medical domains. Within the ALFUS Framework, a three-axis representation—the Contextual Autonomous Capability (CAC) model—was defined to characterize UMSs from the perspectives of requirements, capabilities, and levels of difficulty, complexity, or sophistication. The individually established scores (1–10) along those three axes, namely *Human Independence*, *Environmental Complexity*, and *Mission Complexity* are used to give a standard and straightforward characterization of certain autonomous applications [34].

In this article’s context, a specialized version of the CAC model, Surgical Contextual Autonomous Capability (SCAC), is introduced and customized to the domain of surgical robotics. SCAC extends the Level of Autonomy (LoA, Section 2.1) concept of RAMIS, presented in [10], offering a more detailed classification of autonomous surgical applications. The other foundation of the SCAC model is the Level of Clinical Realism (LoCR) scale for surgical automation, defined in [35] as:**LoCR 1** – *Training tasks with rigid phantoms;***LoCR 2** – *Surgical tasks with simple phantoms;***LoCR 3** – *Surgical tasks with realistic phantoms, but little or no soft-tissue interaction;***LoCR 4** – *Surgical tasks with soft-tissue interaction;***LoCR 5** – *Surgical tasks with soft-tissue topology changes.*

The LoCR concept (Figure 2) can be interpreted as a compound scale, as it includes the complexity of both the environment and the surgical task. Thus, in this work it is decomposed to two individual scales: Level of Environmental Complexity (LoEC, Section 2.2) and Level of Task Complexity (LoTC, Section 2.3). LoA, LoEC, and LoTC are chosen to be the three aspects in the specialized, SCAC model (Figure 3), matching the original concept of the ALFUS Framework. The proposed SCAC model is able the characterize autonomous surgical applications from the perspectives of human independence and difficulty levels regarding the task and the environment. Although this article focuses on surgical subtask automation, our model concerns the whole domain of automation in surgery.

### 2.1. Level of Autonomy

Establishing objective conditions for autonomy has been a historical challenge for the robotics community [39]. First, the Degree of Autonomy (DoA) was introduced in ISO 8373:1994 *Robots and robotic devices – Vocabulary*, but was defined properly only decades later in IEC/TR 60601-4-1: *Medical electrical equipment – Part 4-1: Guidance and interpretation – Medical electrical equipment and medical electrical systems employing a degree of autonomy as “taxonomy based on the properties and capabilities of the medical electrical equipment or medical electrical system related to autonomy”*. IEC/TR 60601-4-1:2017 recommends the parameterization of DoA along four cognition related functions of a system, affecting options of a medical electrical system: *Generate* an option; *Execute* an option; *Monitor* an option; *Select* an option [40]. The LoA concept of RAMIS—originating from the field of autonomous vehicles [41]—was proposed in [10], modified from [9], simplifying DoA to offer a taxonomy to generally assess the development phases of surgical robotics (Figure 4). The proposed 6-grade scale is coherent to the mainstream standardization efforts, and defined as the following:**LoA 0** – *No autonomy*;**LoA 1** – *Robot assistance;***LoA 2** – *Task-level autonomy;***LoA 3** – *Supervised autonomy;***LoA 4** – *High-level autonomy;***LoA 5** – *Full autonomy*.

### 2.2. Level of Environmental Complexity

The common surgical environment can be described more accurately than the broad range of areas included in the ALFUS Framework. The proposed scale is read as follows:**LoEC 1** – *Training phantoms:* made for the training of surgical skills (e.g., hand–eye coordination), no or limited, highly abstract representation of the surgical environment, e.g., wire chaser;**LoEC 2** – *Simple surgical phantoms:* made for certain surgical subtasks, modeling one or few related key features of the real environment, e.g., silicone phantom for pattern cutting;**LoEC 3** – *Rigid, realistic surgical environment:* realistic surgical phantoms or ex/in vivo tissues/organs, little or no soft-tissue interaction, e.g., ex vivo bone for orthopedic procedures;**LoEC 4** – *Soft, realistic surgical environment:* realistic surgical phantoms or ex/in vivo tissues/organs, soft-tissue interaction, e.g., anatomically accurate phantoms for certain procedures or ex vivo environment;**LoEC 5** – *Dynamic, realistic surgical environment:* realistic surgical phantoms or ex/in vivo tissues/organs, soft-tissue topology changes, e.g., in vivo environment with all relevant physiological motions.

### 2.3. Level of Task Complexity

The LoTC represents the Mission Complexity from the ALFUS Framework in the surgical domain. Two components of complexity were compiled into the proposed scale: is it a training or an actual surgical task, and what are the Situation Awareness (SA) requirements of the execution. SA is defined in three levels based on the cognitive understanding of the (past–present–future) environment, and can be categorized into the following classes: spatial (locations), identity (salient objects), temporal, goal, and system awarenesses [42,43]:**Level 1 SA** – *perception of the environment;***Level 2 SA** – *comprehension of the current situation;***Level 3 SA** – *projection of future status.*

Based on the mentioned considerations, the following LoTC scale is proposed:**LoTC 1** – *Simple training tasks:* no or limited, distant representation of surgical task, no or Level 1 SA is required, e.g., peg transfer;**LoTC 2** – *Advanced training tasks:* no or distant representation of surgical task, basic reasoning and Level 2 or 3 SA is required, e.g., peg transfer with swapping rings;**LoTC 3** – *Simple surgical tasks:* no or Level 1 SA is required, e.g., debridement;**LoTC 4** – *Advanced surgical tasks:* Level 2 SA, spatial knowledge and understanding of the scene are required, e.g., suturing;**LoTC 5** – *Complex surgical tasks:* Level 3 SA, clinical and anatomical knowledge are required, e.g., stop acute bleeding.

## 3. Performance Metrics

For the purpose of validation of an autonomous surgical subtask, the most obvious idea would be to compare it with the task execution of human surgeons—even with varying skill levels. It is also viable to compare the performance of the autonomous system to another, already validated one. For both methods, i.e., human–machine and machine–machine comparison, the proper choice of performance metrics that describe how well the subtask is executed is crucial [44].

Since the research domain of surgical subtask automation is still in its infancy, no standard set of metrics is established yet. In the case of surgical skill assessment, the standard practice is to appoint a ground truth for that skill level, e.g., the *number of surgeries performed*, *years of practice*, or manual scoring by an expert surgeon. Then, the metrics measured in the experimental study are correlated to the ground truth to find out which metrics are best related to the surgical skill [44,45]. To extend this methodology of finding the metrics best represent the quality of autonomous surgical subtask execution, there are three possibilities to correlate the metrics of the autonomous execution:(a)to the ground truth utilized in the case of human surgeons;(b)to the metrics from human execution that are found to be correlated with surgical skill;(c)to new ground truth for autonomous execution.

All of these options are found to be quite problematic: in the case of (a), the ground truth metrics (e.g., *years of practice*) could not be interpreted for autonomous surgery; (b) relies on the statement that the same metrics represent the quality of both human and autonomous execution—which is not only not prove, but it is easy to see the opposite e.g., for metrics like *distance traveled* or *jerk*. The only viable solution appears to be option (c), to correlate to the new ground truth for autonomous execution. However, to conduct such a study, a ”population” of autonomous agents would be required, similarly to studies on human surgeons. Unfortunately, assembling a ”population” of autonomous agents, proper for the conduction of a study is not possible, and even the term ”population” is ambiguous in this context.

Although it is not possible to experimentally prove which metrics represent the quality of an autonomous system or the clinical outcome best, a set of metrics can still be recommended based on their various properties. In the following subsections, the area of MIS skill assessment is reviewed briefly in terms of performance metrics, then the candidates for standard performance metrics will be presented, organized by modality. Those metrics are scored and evaluated along different aspects, and finally, a recommendation on standard validation metrics is proposed. Additionally, it is important to note that choosing the fitting metrics to evaluate the performance of the given application greatly depends on the system’s SCAC, the relationship of the metrics to choose and SCAC is also discussed.

### 3.1. Performance Metrics in MIS Skill Assessment

The principle of RAMIS subtask automation is trying to replicate some of the human surgeon’s actions. Thus, the search for metrics to evaluate the performance of an autonomous surgical application should start in the area of MIS skill assessment. Finding the metrics that best correlate with the surgical skills is far from trivial, and already has an extensive literature [44,45]. For example, one could think that *mortality rate* after the surgery would somehow relate to the surgical skills; the better the surgeon the lower the *mortality rate* would be. However, there are a number of different factors contributing to this metric. For instance, a beginner surgeon may not undertake the operation of patients with poor health conditions, and does low-risk surgeries instead, resulting in low *mortality rate*. In contrast, an expert surgeon may be more willing to undertake high-risk interventions, but that could result in a higher *mortality rate*, despite how well the interventions are performed [46].

One of the most widely used standard surgical skill assessment techniques is the Global Evaluative Assessment of Robotic Skills in Endoscopy (GEARS-E) [47], in which the following aspects of execution are scored: *depth perception*; *bimanual dexterity*; *efficiency*; *tissue handling*; *autonomy*; *endoscope control*. A quite similar and also prevalent technique is the Robotic Objective Structured Assessments of Technical Skills (R-OSATS) [48], scoring *depth perception/accuracy of movements*; *force/tissue handling*; *dexterity*; *efficiency of movements*. Both methods utilize manual scoring 1 to 5 using the Likert scale [49], thus being subjective, making it hard to be used in the validation of autonomous applications. Raison et al. [50] compiled their surgical simulation training study using a custom score set divided to general scores: *time to complete*; *economy of motion*; *master working space*; *instruments out of view*; *excessive force*; *instrument collision*, and task specific scores: *blood loss*; *broken vessels*; *misapplied energy time*; *missed target*; *dropped instrument*.

The above-mentioned metrics all represent the technical skills of surgery. However, the outcome of the surgical procedure is also dependent on the non-technical skills of the surgeon, and those cannot be interpreted in the case of an autonomous application [44,51]. Those skills are typically rated using a questionnaire filled by the subject, like the NASA Task Load Index (NASA-TLX) [52], evaluating *mental demand*; *physical demand*; *temporal demand*; *performance*; *effort*; *frustration*. Non-technical skills are not discussed in more detail due to the lack of their usefulness in automation unless any adverse event occurs.

### 3.2. Metrics by Modality

In the followings, various performance metrics are overviewed, and their usability is discussed. Those metrics are collected from the literature of surgical subtask automation, and also from the fields of autonomous robotics, autonomous vehicles, and surgical skill assessment, and are organized into subsections by the modality of the measured values [44,45,53].

#### 3.2.1. Temporal Metrics

*Completion time* is one of the most commonly used metric both in surgical skill assessment [44,45] and in surgical subtask automation [15,19,24,27,29,54,55]. It characterizes the whole execution of the subtask, and could be defined in a number of ways, e.g., in the case of peg transfer, *completion time* can be taken as the average of each individual transfer, or as the average for whole subtask executions. For humans, temporal metrics tend to correlate with skill level or give a measure of hesitation. In the case of automation, those connect more loosely to the quality of the execution. However, as the lower time requirement of surgical interventions is beneficial—for both the patient, surgeon, or the hospital—near-human, or even superhuman *completion time* is still an important factor in automation. For instance, Sen et al. [17] measured *completion time* for autonomous suturing and compared it to manual executions from the JHU-ISI Gesture and Skill Assessment Working Set (JIGSAWS) database [56]. Additionally, Ginesi et al. [25] validated their autonomous task planning algorithm for peg transfer by measuring the *task planning time* in different scenarios.

Temporal metrics could also be used to evaluate the elements of the whole system. *Time to compute* is one of the most important metrics in the field of computer science and could describe e.g., perception algorithms, trajectory generation, or planning in autonomous systems [25]. *Completion time* and *time to compute* become extremely useful when working on benchmarks, then it can be quite a strong factor of comparison between different solutions, offering inter-comparability for different research groups.

Time is also critical in the reaction to adverse events. Current research efforts in the area of RAMIS aim at LoA 2, where the surgeon’s supervision is essential, as they need to recognize and solve adverse events. The *reaction time* of autonomous surgical systems becomes important—if not crucial—at LoA 3+, where the autonomous system has to recognize and react to unexpected events either by solving the emergency autonomously or by sending a handover request to the human surgeon.

#### 3.2.2. Outcome Metrics

Outcome metrics assess the end result of the whole procedure—or in this case subtask—or its elements individually, ignoring the way it is performed completely. Such metrics are e.g., *number of errors*, *quality of the outcome*, and *success rate*. *Success rate* is probably the most universal and easy to measure, thus utilized frequently in surgical subtask automation [15,19,27,29,31,54,55,57]. McKinley et al. [57] evaluated autonomous tumor resection by end-to-end *success rate*, while Hwang et al. [27] defined it for the peg transfer training exercise as the percentile value of the ratio of *Success/Attempts* for each individual transfer. Attanasio et al. [21] utilized *visible area* as outcome metric in autonomous retraction. Nguyen et al. [30] measured the accuracy of pattern cutting next to autonomous tensioning. In the study of Shademan et al. [14], autonomous end-to-end anastomosis is presented and validated in vivo on porcine, where *number of sutures*, *number of suturing mistakes*, *leak pressure*, *luminal diameter reduction*, *weight at surgery*, and *weight at sacrifice* are measured and compared to manual execution.

It is important to note that outcome metrics are highly task-specific, thus, making any comparison between different subtasks is very difficult. On the other hand, the implementation of those metrics requires less effort than most of the others.

#### 3.2.3. Motion-Based Metrics

Motion-based metrics utilize the position (and sometimes orientation) of the surgical instrument, the surgeon’s hands, or other tracked objects, as a trajectory or motion profile [44,50,58,59,60]. Some of the simplest ones are *distance traveled*, *economy of motion*, and *number of movements*, but it is also common to use hidden Markov models or other machine learning algorithms to compare the movement patterns to expert surgeons. Those metrics offer an objective way to assess the skill of human surgeons, hence, those give a measure of hesitation, dexterity, and motor skills. However, their usefulness is limited in the field of surgical subtask automation. For example, *distance traveled* could be decreased or increased programmatically without major effects on the quality of execution. Additionally, for the more advanced metrics, comparing the motion pattern to experts could also be misleading in the case of autonomy; the motion patterns of human experts contain the restraints and signature of human anatomy, but an autonomous robot could possibly execute the same task through different—even more beneficial or optimized—trajectories.

#### 3.2.4. Velocity and Acceleration Metrics

Velocity and acceleration metrics are calculated as the first and second derivatives of the motion profiles mentioned regarding motion-based metrics [59,61]. These are widespread in surgical skill assessment, to name some without being exhaustive: *peak speed*, *normalized speed*, *number of changes in velocity over time*, *number of accelerations and decelerations*, *mean acceleration*, or the *integral of the acceleration vector*. These are related to similar traits as the motion-based metrics, and thus their usefulness in automation is questionable.

#### 3.2.5. Jerk Metrics

Jerk is a metric deriving from the third derivative of the motion profile. As it has a quite broad literature from surgical skill assessment [53,59,62] to the diagnostics of neurodegenerative diseases [63,64], it deserves to be mentioned separately. Jerk metrics also tell us about motor skills; motion patterns are usually become smoother by practice. However, its usage in automation is quite insignificant, since it is a highly human-specific metric.

#### 3.2.6. Force-Based Metrics

The amount of force applied to the tissues is a significant characteristic of surgery; it is important not to cause damage by excessive force, but it is also crucial to provide enough tension e.g., during tightening a knot [65,66]. Additionally, Trejos et al. [53] have shown in their experimental study that force-based metrics correlate better with experience levels than temporal metrics in manual MIS. Unfortunately, sensorizing MIS or RAMIS instruments is still very challenging due to the small dimensions or sterilization requirements, and thus, the usage of force-based metrics is not prevalent in skill assessment or in the evaluation of automation. However, currently, there are several solutions to measure or estimate forces and torques on the shaft or the tip of the instruments [67,68], the phantoms, or even to measure the applied grasping force [69,70]. Using such devices, the metrics of the force applied, e.g., *grasp maximum*, *grasp integral*, *cartesian maximum*, or *cartesian integral* could be included to the validation of surgical subtask automation. Such an example can be seen in the work of Osa et al. [71], where the applied force was measured during autonomous thread tightening.

#### 3.2.7. Accuracy Metrics

Accuracy is a minor problem in surgical skill assessment, but it is extremely important in many areas of automation. Accuracy metrics usually characterize one or few subcomponents or aspects of the autonomous system, such as the *positioning accuracy* of the utilized robots and low-level controllers, *accuracy of hand–eye registration*, *pose estimation*, or *object detection*. The current best practice is to measure the accuracy of the system’s components in surgical subtask automation in order to validate the application. For instance, Lu et al. [72] validated their knot-tying application by measuring the *tracking error* on the instruments. Additionally, Lu et al. [15] measured the *tracking error* of the grasping point in their study on autonomous suture grasping. Sen et al. [17] measured the *accuracy of needle detection* in their study on autonomous suturing. Elek et al. [73] measured *depth error of the camera*, *positioning error*, and *accuracy of dissection profile extraction* among automation metrics in their autonomous blunt dissection study. Besides validation, accuracy metrics could also help localize problems during the implementation phase.

Although measuring metrics such as the *robot positioning accuracy*, the *accuracy of hand–eye registration*, the *instrument tracking error*, or the *accuracy of pose estimation* can be highly beneficial, especially during the implementation phase, those properties and errors are all contributing to the *end-to-end positioning accuracy*. Thus, for the purpose of validation, those measurements could be substituted by testing and measuring *end-to-end positioning accuracy*, or *application accuracy*, that can even tell more about the accuracy of the whole system, than the accuracy of its components [74].

The *application accuracy* could be measured most precisely by incorporating a high-precision external system, such as industrial robots or tracking systems (visual or electromagnetic). This method is highly recommended, however, in some cases, this is not a viable option due to the high cost of such devices, and low fidelity methods need to be utilized. Pedram et al. [75] measured the *error of pose estimation* of the needle, then used this pose estimation to measure the application accuracy of the needle in their study on autonomous suturing. Seita et al. [19] used the following method to fine-tune the end-to-end positioning of their system for autonomous debridement: the tool was sent to points on some kind of grid, using positions from their computer vision solution, then the tool position was adjusted manually to the desired position on the grid, and from the kinematic data of the robot, the *application accuracy* was calculated. As this method, especially by utilizing a grid with physical constraints to help accurate manual positioning (e.g., holes or dips for the tool endpoint), offers a simple yet accurate method for the measurement of the *application accuracy*. The utilization of such methods is also viable for the validation of autonomous systems in RAMIS but only recommended if the errors of the ground truth system are known and error propagation is taken into account.

### 3.3. Conclusions on Performance Metrics

The main scope of this article is to introduce a criteria set for the evaluation and validation of subtask-level automation in RAMIS. The most characteristic and meaningful metrics are compiled into Table 1, and scored for usability in the targeted area of the assessment of systems at LoA 2 in three aspects: *Task Independency*, *Relevance with Quality*, and *Clinical Importance*. The metrics with the best overall scores are highlighted with dark gray, those are highly recommended for the validation of autonomous subtask execution in RAMIS. The metrics highlighted with light gray are found to be moderately useful, the utilization of those could be considered in a number of subtasks as well.

Based on the overall scores, outcome and accuracy metrics (*accuracy of object detection*, *accuracy of pose estimation*, and *application accuracy*) were found to be the best among all for the validation of autonomous surgical systems, in general. It is important to note that accuracy metrics perform slightly better in task independency, since, unlike outcome metrics, those offer inter-comparability between different tasks. Temporal and force-based metrics also received good overall scores, with *reaction time* and *Cartesian force* in the “highly recommended” category. Although the measurement of force-based metrics requires additional sensors, unlike other metrics, those metrics tell a lot about the quality of execution and also the utilized force is very important clinically. Motion-based, velocity, acceleration, and jerk metrics received relatively low scores, the usability for the performance assessment of autonomous surgical subtasks were found questionable.

The choice of metrics greatly depends on the certain subtask and the experimental setup. Thus, a flowchart is supplied that can be used to compile a list of metrics for the validation of the given autonomous surgical application (Figure 5). The list of metrics depends on, among the others, the type of benchmarking environment, the handing of soft tissues, or the LoEC and LoA of the system. For example, for the validation of an instrument pose estimation algorithm for surgical videos using a benchmark dataset requires *completion time accuracy of object detection* and *accuracy of pose estimation*, while the validation of a system performing autonomous peg transfer should to be validated using *completion time*, *application accuracy*, *success rate*, *accuracy of pose estimation* and *accuracy of object detection*. Important to note that *accuracy of pose estimation*, and *accuracy of object detection* are suggested to be added to the list in any case, since those can be used for the validation of almost any application in the field of surgical subtask automation, and also received the highest overall scores.

## 4. Benchmarking Techniques

The recommended metrics allow the comparison of different autonomous applications. Additionally, testing and evaluating autonomous applications in a benchmark environment—a standardized problem or test that serves as a basis for evaluation or comparison—offers an even more solid basis for both human–machine and machine–machine comparisons [76]. Currently, the usage of benchmarking techniques is not prevalent in the field of surgical subtask automation, still it is used intensively in the area of autonomous driving. The development of autonomous vehicles is considered analogous to surgical subtask automation due to the high complexity of the environment and the presence of potentially life-threatening risks [77].

As the development of autonomous subtask execution in surgery progresses and gets closer to clinical use, the need for evidence on the autonomous system’s capabilities will rise. In [78], Fiorini states the followings: *“Benchmarks will be the key to build a solid confidence on the robots’ capabilities to understand and react to clinical situations as a human surgeon would do, and the process of medical certification for surgical robots will need to be developed.”* Next to low LoEC it is relatively convenient to create benchmarks, but unfortunately, as LoEC increases, meaning realistic or even dynamic environments, the development of benchmarks will be considerably more difficult.

In the research of self-driving, benchmarking techniques can be compiled into three categories: benchmark datasets to test system components like object detection accuracy [79,80,81,82]; standard simulated environments like the scenarios developed for the CARLA Car Simulator [83,84]; physical test towns like mCity [85].

Autonomous surgical subtasks could also be evaluated using the mentioned types of benchmarks. Benchmark datasets for instrument segmentation are already available within the yearly MICCAI Challenges, including ground truth for training and evaluation of accuracy [86,87]. Among the surgical simulators, the Asynchronous Multi-Body Framework (AMBF) [88] has to be emphasized, as it is open-source, offers da Vinci and Raven robot arms, and supports the simulation of deformable objects. Moreover, AMBF contains a built-in peg transfer board that could already serve as a benchmark. Important to note the *2021 AccelNet Surgical Robotics Challenge*, also based on the AMBF simulator, offers a simulated setup to develop autonomous suturing applications [89].

Benchmarking physical setups presents a bigger challenge to implement [90]. Thanks to 3D printing technology it is possible to create and distribute some standard physical benchmarks, such as training phantoms or rigid surgical phantoms. The peg transfer board presented in [27], whose design files are freely available, is a perfect candidate for a standard physical benchmark. By defining a standard execution of the exercise and standard evaluation metrics a simple benchmark could be compiled to measure and compare the performance of different autonomous algorithms.

The utilization and spread of benchmarks alongside the standard performance metrics in RAMIS subtask automation would potentially make the performance of different autonomous systems much more comparable. Since a number of challenges offer benchmark datasets, even standard simulated environments, the authors highly recommend the usage of these as is. In terms of physical benchmarks, 3D printing technology makes the sharing and reproducing of phantom environments and other objects easier and cheaper than ever [36], and hopefully the practice of using 3D printable surgical phantoms will soon spread in the research community.

## 5. Human–Machine Interface Quality

The quality of the HMI has an important role in the clinical usability of autonomous surgical systems, especially between LoA 1 and LoA 4, where the human surgeon and the autonomous system perform the surgical intervention together. In the field of RAMIS subtask automation, it is quite uncommon to test and validate the HMI of the autonomous system. However, in related research fields, such as self-driving vehicles or image-guided surgery, the HMI is validated much more frequently [91,92]. Usually, the quality of the HMI is assessed using the performance metrics seen in Section 3, like *time to complete* or *success rate*; the aim of those tests is to assess the performance of the human surgeon and the autonomous system together.

Another important aspect of HMI quality is the system’s hand-over capability, especially with LoA 2–4 systems. During a hand-over, as it is described in Section 3.2.1, time is the most crucial factor. Firstly, the autonomous system should recognize the adverse event and have to initialize a hand-over request to the surgeon as soon as possible, then the autonomous system has to yield the control safely. Secondly, in the case of surgeon-initiated hand-overs the system has to yield the control to the surgeon again, safely and with the lowest possible delay [93].

The assessment of general HMI quality and hand-over capabilities is rarely found in surgical subtask automation-related studies currently. However, such tests could enhance the clinical relevance, and also could improve the trust in those systems, by the public and the relevant authorities.

## 6. Robustness

Robustness is defined in a number of ways [94], the followings are the best fitting to surgical subtask automation:*The ability...to react appropriately to abnormal circumstances (i.e., circumstances “outside of specifications“). [A system] may be correct without being robust.* [95];*Insensitivity against small deviations in the assumptions* [96];*The degree to which a system is insensitive to effects that are not considered in the design* [97].

Reaching higher LoA, LoEC, and LoTC, the robustness of autonomous surgical systems becomes crucial. It is conclusive that LoA cannot be increased without higher robustness, hence unexpected events may result in unwanted human handover requests, potentially causing a dangerous scenario. The surgeon’s SA might decrease during autonomous execution, affecting handover performance negatively. Furthermore, as LoEC and LoTC are increasing, the number of uncertainties and not considered circumstances also rises.

To measure the robustness of an autonomous system, additional methods should be utilized, the performance of the system should also be tested next to unexpected events. In the case of the autonomous training task, it might be relatively simple to even manually generate such events (e.g., accidentally dropping grasped objects). Another example of robustness testing can be seen in the work of Elek et al. [73], where the performance of the perception algorithm was measured on different textures. However, in the case of higher complexity, the utilization of automatic robustness testing software may be necessary [98].

It is also common in deep learning to add noise to the input to increase robustness and avoid overfitting, like in the autonomous soft-tissue tensioning application by Nguyen et al. [30].

## 7. Legal Questions and Ethics

During the academic research phase, legal and ethical aspects of an autonomous surgical application or system are usually considered less. However, at the point when development approaches clinical usage, those aspects become critical. Since autonomous surgical systems could potentially endanger the life of the patient, the introduction of new standards and regulations in the field can be extremely difficult and elaborate. The availability of best practices for the validation of such systems could support those processes.

In the field of automation, liability is usually a prickly question, but in general, as LoA increases, the liability in RAMIS is gradually shifting from the surgeon to the manufacturer of the system. The regulating authorities, such as the European Commission in the case of the European Union, are to protect citizens from harm caused by, in this case, autonomous surgical applications. Thus, in order to commercialize such solutions, the manufacturer needs to adequately demonstrate that the autonomous system is capable of performing the intervention with equal or better performance as a human surgeon would do [78]. A proper characterization model and standard evaluation metrics would probably be quite useful in the procedure of legal approval.

The effect of automation on the surgeons’ performance is also a significant concern. The utilization of autonomous functions in surgery may increase the reliance on them and can lead to a decrease of skills for human surgeons. This lack of skills can be crucial and even may risk the life of the patient in cases when the autonomous system fails and the human surgeons need to take over—especially if such failure is infrequent.

The definitions and safety requirements of surgical robotics are established in the standard IEC 80601-2-77 [40]. The future standardization of the SCAC model, metrics, benchmarks, or even autonomous surgical applications could be initiated through the following organizations:International Organization for Standardization (ISO);International Electrotechnical Commission (IEC);Institute of Electrical and Electronics Engineers (IEEE)Strategic Advisory Group of Experts (SAGE), advising World Health Organization (WHO);European Society of Surgery (E.S.S.) in Europe;Food and Drug Administration (FDA) in the USA.

In addition to the medical ethics, like the Hippocratic oath of *”do no harm”*, Artificial Intelligence (AI) ethics are also touching on autonomous surgery, such as the FAST Track Principles (Fairness, Accountability, Sustainability, and Transparency), and should be considered during development. Ensuring fairness, like avoiding algorithmic or statistical biases, is essential, since in the case of autonomous surgery those biases may lead to fatal consequences. Additionally, for example, in the case of deep learning methods, transparency could not be ensured; although the result of the network shows good accuracy, what it learned could not be interpreted by humans, and no one could predict its output from inputs yet unseen by the network [99].

Concepts such as sustainability, from *roboethics*, a research field, concentrating on ethics in robotics [100], should also be applied to the development of autonomous surgical systems. Moreover, regulations on safety and privacy must be followed, like the Medical Devices Regulation (EU) 2017/745 (MDR) and the General Data Protection Regulation (EU) 2016/679 (GDPR) within the European Union. Furthermore, the contents of the recently published standard IEEE 7007^™^ – *Ontological Standard for Ethically Driven Robotics and Automation Systems* is also going to be of critical significance in the development of surgical subtask automation.

## 8. Conclusions and Discussion

In this article, standard evaluation methods, metrics, and benchmarking techniques were proposed for performance evaluation and validation of systems executing surgical subtasks autonomously. A three-axis SCAC model is proposed to represent more detailed characterization of the autonomous capabilities in RAMIS and also in the wider field of surgery. This SCAC model uses the five and six grade scales of LoA, LoEC, and LoTC to represent autonomous surgical systems from a broad view.

Based on the review of literature, a set of performance metrics are presented and grouped by modality. After scoring the metrics by the aspect of usability of RAMIS subtask automation, the metrics were ranked, their properties were discussed. In the field of surgical subtask automation the most widely used, and also the most meaningful metrics are the outcome and accuracy metrics. Outcome metrics like *success rate* are easy to implement, but—especially as SCAC increases—specific outcome metrics can also be defined to the given subtask. Accuracy metrics are also quite useful to validate components of the autonomous system, as *accuracy of pose estimation* and *accuracy of object detection*, are among the ones that could and should be used for the validation of almost any autonomous surgical system, or their subsystems. The measurement of temporal metrics is very common, but as long as it is similar to human execution, it is less informative, except in the case of benchmarks, where those become a good basis of comparison. Currently, in the field, the utilization of force-based metrics is quite uncommon, but as LoEC increases, the validation of delicate tissue handling will probably become a critical issue.

A recommendation for the universal list of metrics cannot be given, since the metrics best represent the quality of the autonomous execution greatly depend on the task and the validation environment. Instead, a methodology of choosing metrics for the validation of certain autonomous applications was also proposed and illustrated in a flowchart (Figure 5).

A review of current and a proposal for further benchmarking techniques are presented for surgical subtask automation. Benchmarks are going to have a significant role in the future, during the introduction of subtask-level autonomy to the clinical practice by supporting the authorities’ and the patients’ trust in the autonomous systems. At present, a significant portion of research involves the automation of the peg transfer training task, hence, it serves as an adequate model to act as a foundation for surgical subtask automation. Additionally, the design files necessary for the 3D printing of the board and the pegs are available online [27]. In terms of benchmark datasets and simulations, the materials for a number of challenges are available, and even when the challenge is over, it could still serve as a good benchmark for future research projects.

The matter of HMI quality, robustness, and legal and ethical questions were also discussed in brief. One of the most serious concerns regarding autonomous surgical systems—below LoA 5, *full autonomy*—is how a hand-over could be performed in the case of an emergency or malfunction. This concern is further strengthened by the trends in other areas, like self-driving vehicles, where automation has led to increasing reliance on those autonomous systems by the human operator.

The mentioned numerous concerns and fears related to autonomous surgical systems put an increasing need on the research community to perform through validation and testing on their developed applications. The proposed methodologies and recommendations could help the community to quantitatively and soundly measure the quality of the autonomous executions, and to provide a ground to compare the results of various research groups. At the point, when surgical subtask automation will start to break into the clinical practice, the proposed methodologies could also be used as the basis of emerging standards.

## Figures and Tables

**Figure 1 sensors-22-02501-f001:**
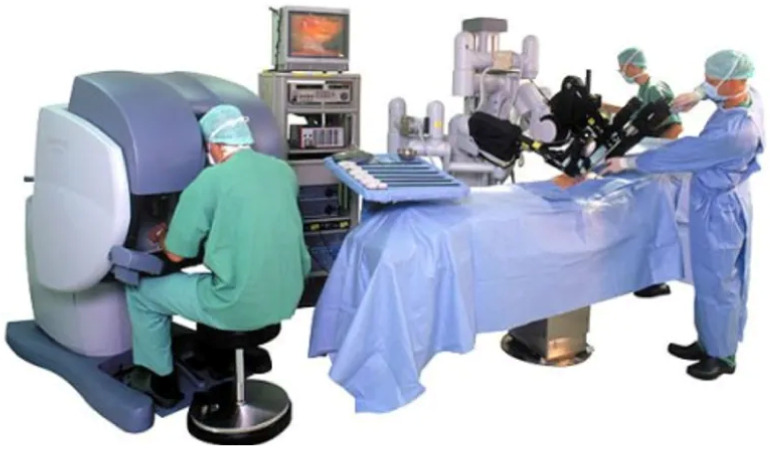
The da Vinci classic teleoperated surgical system, which is still a popular platform for RAMIS research, due to the DVRK extension. *Image credit: Intuitive Surgical*.

**Figure 2 sensors-22-02501-f002:**
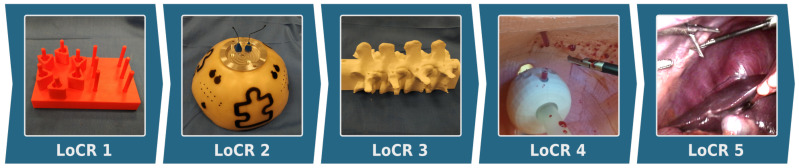
The Level of Clinical Relevance (LoCR) scale for Robot-Assisted Minimally Invasive Surgery (RAMIS) with examples. *LoCR 1:* 3D printed board for the peg transfer training task, designed by Hwang et al. [27]; *LoCR 2:* Fundamentals of Robotic Surgery (FRS) training dome; *LoCR 3:* 3D printed bone phantom for drilling tasks; *LoCR 4:* anatomically relevant silicone pelvis phantom [36]; *LoCR 5:* in vivo human surgical environment [37].

**Figure 3 sensors-22-02501-f003:**
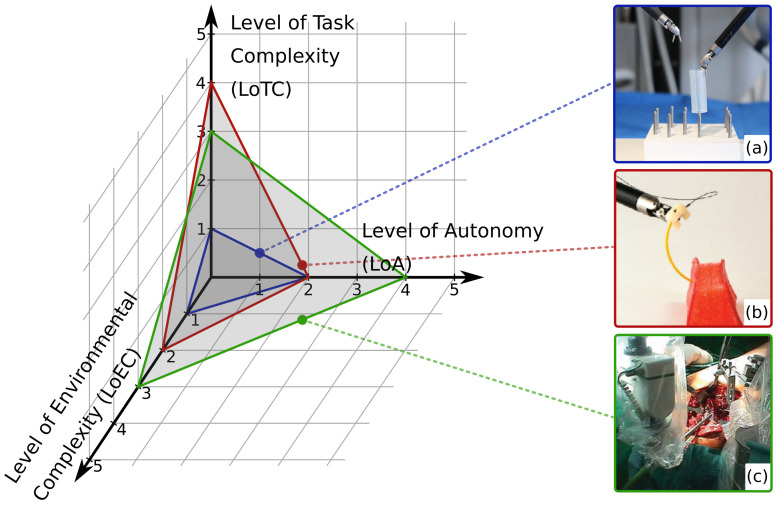
The three-axis model of Surgical Contextual Autonomous Capability (SCAC). The *x*, *y*, and *z* axes represent the key characterizing aspects of autonomous surgical applications: Level of Autonomy (LoA), Level of Environmental Complexity (LoEC), and Level of Task Complexity ((LoTC), respectively. The characterization of three example applications are illustrated: (**a**) *autonomous peg transfer surgical training exercise* [22] (grading LoA 2; LoEC 1; LoTC 1); (**b**) *autonomous multi-throw multilateral surgical suturing* [17] (LoA 2; LoEC 2; LoTC 3); (**c**) *autonomous bone drilling for total hip replacement surgery*, performed by the ROBODOC system (CUREXO INC., Seoul, Korea) [38] (grading LoA 4; LoEC 3; LoTC 3).

**Figure 4 sensors-22-02501-f004:**
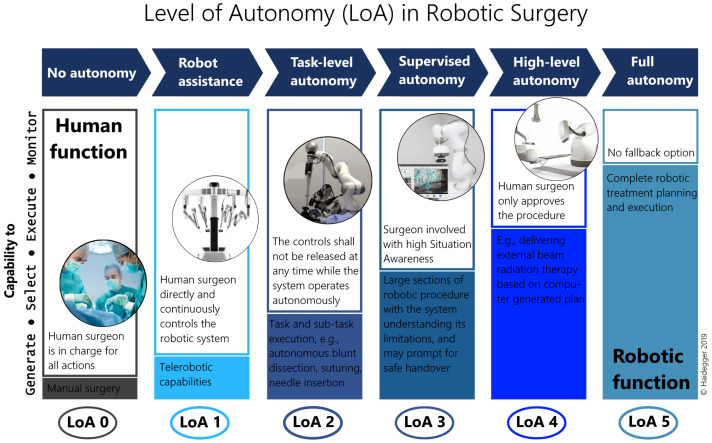
The Level of Autonomy (LoA) concept for Robot-Assisted Minimally Invasive Surgery (RAMIS) proposed by Haidegger [10] with a scale 0–5; from no autonomy to full autonomy.

**Figure 5 sensors-22-02501-f005:**
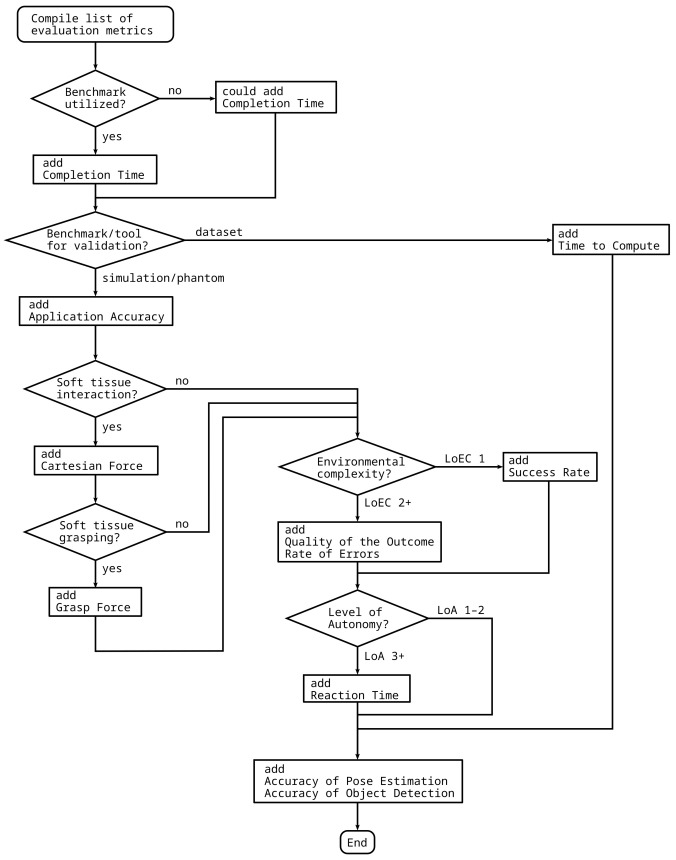
Flowchart to compile a list of performance metrics for the validation of different autonomous applications. The proposed method requires different properties of the autonomous surgical application, the task and the validation environment as input, and outputs the list of recommended validation metrics.

**Table 1 sensors-22-02501-t001:** Characteristic performance evaluation and comparison metrics for RAMIS subtask automation, grouped by modality. The metrics are scored in three aspects on a scale of 1–3: *Task Independency* (1—usable only for specific tasks; 2—can be used for any task, but not inter-comparable; 3—can be used for any task, different tasks are comparable); *Relevance with Quality* of task execution (1—irrelevant; 2—relevant, but cannot correlate with quality; 3—relevant and correlates with the quality of task completion); *Clinical Importance* (1—not important; 2—moderately important; 3—very important). The scores are summed in the last column for each metric and the ones with the best scores are highlighted with light gray (6–7) and dark gray (8–9).

Modality	Metric	Task	Relevance	Clinical	Overall
		Independency	with Quality	Importance	Score
Temporal	Completion Time	2	2	2	6
	Time to Compute	2	2	2	6
	Reaction Time	3	2	3	8
Outcome	Rate of Errors	2	3	3	8
	Quality of the Outcome	2	3	3	8
	Success Rate	2	3	3	8
Motion-based	Distance Traveled	2	2	1	5
	Economy of Motion	2	2	1	5
	Number of Movements	2	2	1	5
Velocity and Acc.	Peak Speed	2	1	1	4
	Number of Accelerations	2	1	1	4
	Mean Acceleration	2	1	1	4
Jerk	Jerk	3	1	1	5
Force-based	Grasp Force	1	3	3	7
	Cartesian Force	2	3	3	8
Accuracy	Accuracy of Pose Estimation	3	3	3	9
	Accuracy of Object Detection	3	3	3	9
	Application Accuracy	2	3	3	8

## Data Availability

Not applicable.

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
