# Peer review of "Performance and Capability Assessment in Surgical Subtask Automation"

_sensors, 2022, doi:10.3390/s22072501_

Round 1
Reviewer 1 Report
This paper presents some useful information to help guide the analysis or validation of autonomy or partial autonomy in surgical robotics. The paper first introduces some categorizations, including definitions of Level of Environmental Complexity (LoEC) and Level of Task Complexity (LoTC), in addition to the better known Level of Autonomy (LoA). The LoA presented by the authors in Fig. 3 differs slightly from the more frequently cited one in [9], especially for levels 3 and 4. Based on [10], it seems that the rationale for this variation is to "distinguish the required level of human supervision" and Fig. 3 clearly indicates this information.
The paper next presents performance metrics (Section 3). The introduction to this section is rather bleak, presenting three possibilities for metrics and then indicating that all of them are "quite problematic". After reading that introduction, I expected the section to end, whereas in reality, the authors actually present a lot of useful metrics. Perhaps this introduction can be revised to better foreshadow the included content.
Section 4 presents some emerging benchmarks, which are still somewhat lacking in the field. Note that there is a recent RAL paper about the AccelNet Surgical Robotics Challenge (Munawar et al., "An Open Simulation Environment for Learning and Practice of Robot-Assisted Surgical Suturing").
I especially appreciate Section 5 (HMI) which, although brief, highlights an important and often overlooked aspect of these systems. Sections 6 and 7 also briefly present Robustness and legal/ethical questions. In the latter case, the authors did not mention one significant concern with these systems, which is that increasing reliance on them (and their autonomous capabilities) may decrease the surgical skills needed for humans to take over when they fail, especially if such failure is infrequent.
Although the article is understandable, it needs significant editing for the language. Most of the issues are with the English grammar, but beyond that, in a few places the style is more like an informal conversation than a scientific text. I note a few typos and wording issues below, but there are many more:
Line 127: "ortopedic" should be "orthopedic"
Figure 4 caption: "&itenigicser2016" probably is meant to be a citation
Line 170: "that found to be correlating" should be "that are found to be correlated"
Line 173: "these options found" --> "these options are found"
Line 179: "no analogous". Maybe "nothing analogous" would be better.
Line 181: "whitch metrics represents" --> "which metrics represent" (2 changes)
Line 183: "Also, important to note" --> "Also, it is important to note"
Line 197: "For Instance" --> "For instance"
Line 209: "making hard to used in" --> "making it hard to be used in"
Line 221: "advert event". Should this be "adverse event"?
Line 304: "Trejos et al. [51] shown" -- "Trejos et al. [51] have shown"
Line 308: "it is not prevalent nor in skill assessment nor in the evaluation of automation". I would change to "it is not prevalent in skill assessment or in the evaluation of automation"
Line 341: "dirty" methods -- could you find a better phrase? Perhaps "lower fidelity"?
Line 348: "accuracyr"
Line 429: "time to complete of success rate" -- I think "of" should be "or"
Line 435: "it have to yield" --> "it has to yield" (also "have" to "has" in line 436)
Line 524: "even the challenge is over" --> "even when the challenge is over"
Reviewer 2 Report
Dear Authors,
Please find attached my comments and suggestions to be addressed before publishing the work.
Kind regards,

Reviewer 3 Report
This paper presents a characterization model for surgical automation. A three-axis SCAC model is proposed to represent a more detailed characterization of autonomous capabilities in surgery. The best metrics for evaluating and comparing automated surgical subtasks are reviewed.
Round 2
Reviewer 2 Report
The paper has improved.
But, I think the English language still needs to be checked.
Good luck
kind regards